# A Low-Cost Ecofriendly Oxidation Process to Manufacture High-Performance Polymeric Biosurfactants Derived from Municipal Biowaste

**DOI:** 10.3390/polym16111479

**Published:** 2024-05-23

**Authors:** Elio Padoan, Francesco Contillo, Matteo Marafante, Enzo Montoneri, Matteo Francavilla, Silvia Berto, Andrea Baglieri

**Affiliations:** 1Dipartimento di Scienze Agrarie, Forestali e Alimentari, Università di Torino, 10095 Grugliasco, Italy; elio.padoan@unito.it; 2STAR Integrated Research Unit, Università di Foggia, 71121 Foggia, Italy; francesco.contillo@unifg.it (F.C.); matteo.francavilla@unifg.it (M.F.); 3Dipartimento di Chimica, Università di Torino, 10125 Torino, Italy; matteo.marafante@unito.it (M.M.); silvia.berto@unito.it (S.B.); 4Dipartimento di Scienze delle Produzioni Agrarie e Alimentari, Università di Catania, Via S. Sofia 98, 95123 Catania, Italy; abaglie@unict.it

**Keywords:** oxidation, autocatalysis, municipal biowaste, biosurfactants, biopolymers, antifungal agents, anaerobic fermentation, composts

## Abstract

Biosurfactants account for about 12% of the global value of the surfactant market, which is currently dominated by synthetic surfactants obtained from fossil sources. Yet, the production of biosurfactants from renewable feedstock is bound to increase, driven by the increasing pressure from both society and governments for chemistry-based industries to become more ecofriendly and economically sustainable. A photo-chemical oxidation process is reported here, yielding new biosurfactants from urban biowaste in water that perform as a solvent and terminal oxidant reagent at room temperature without the addition of conventional oxidants and catalysts. Products with 200–500 kDa molecular weight are obtained. They lower the surface tension of water down to 34 mN/m at 0.5–2 g/L concentration. The estimated cost is rather low (0.1–1.5 EUR/kg), which is competitive with the cost of synthetic surfactants but much lower than the cost of the best-performing bacterial surfactants. For the implementation of the photo-chemical oxidation process at the industrial level, the results suggest that the new biosurfactants obtained in the present work may not reach the performance level of the best-performing bacterial surfactants capable of lowering the surface tension of water down to 28 mN/m. Yet, the biosurfactants produced by the photo-chemical process have a greater chance of being marketed on large scales.

## 1. Introduction

Surfactants are used as active principles in formulations for the manufacture of a wide variety of consumers’ finished products, as well as auxiliaries in many chemical and environmental technologies. Overall, about 60% of EU chemical sales include products containing surfactants [1]. The global surfactant market size is estimated at about 48 billion USD/yr. and is expected to reach 66 billion USD/yr. in 2027 [2] with an estimated production of 13–17 Mt/yr. [3]. Surfactants act as cleaning, wetting, dispersing, emulsifying, foaming, and anti-foaming agents in many practical applications and products, including detergents, personal products, food, pharmaceuticals, plant protection, agriculture, and textiles [4,5].

A key driver of the surfactant market is the growing demand for sustainable and eco-friendly products. With increasing awareness about environmental issues and the need to reduce carbon footprints, consumers are looking for products that are made using renewable resources and do not harm the environment. Surfactants derived from natural sources, such as vegetable oils and sugar, have gained popularity as they are biodegradable, non-toxic, and do not contribute to water pollution. Manufacturers are investing in research and development to produce surfactants that are both sustainable and effective, driving the growth of the market. Relevantly, the biosurfactant market size is currently at the level of USD 8 billion and is expected to reach USD 14 billion by 2023, with a USD 3 billion share of Rhamnolipids sales [6].

Rhamnolipids and sophorolipids are high-performance biosurfactants produced by microorganisms, e.g., *Pseudomonas aeruginosa* and *Rhodococcus erythropolis*, respectively [7]. Natural surface-active compounds, including the above microbial biosurfactants and products extracted from plants and vegetable residues, exhibit very low surface tension values (28–45 mN/m) at 0.4–2 g/L concentration in water, which makes their performance in real operational conditions very competitive with synthetic surfactants from fossil sources. However, the cost of the surfactants from renewable resources ranges from tens to hundreds EUR/kg [7], compared to 1–6 EUR/kg for most commercial synthetic or natural surfactants from petrochemical or natural feedstock [5,8,9]. Yet, in spite of the high cost, the exceptionally low 28 mN/m achievable surface tension and 0.8 g/L operational concentration make rhamnolipids sharing a significantly high USD 3 billion [6] portion of the global surfactant market turnover [2,3,4,5].

In the above context, several papers have been published [10,11,12,13,14,15,16,17] on the properties and performance of new soluble polymeric biosurfactants obtained by hydrolysis and oxidation of collected and fermented municipal biowaste (MBW).

The new polymeric biosurfactants are a heterogeneous mixture of molecules with a molecular weight in the 5 to >750 kDa range and different chemical compositions, constituted by aliphatic C chains and aromatic lignin-like C moieties, and various functional acidic and basic groups of different strengths bonded to several mineral elements, such as Na, K, Ca, Mg, Al, Fe, and Si in 0.2–9 w/w % and Cu, Ni, and Zn in 20–180 ppm content [17]. These features are memories of the chemical moieties and macro-molecularity of the pristine MBW polysaccharide, protein, and lignocellulosic approximates from which the polymeric new biosurfactants are derived [16].

The product obtained by hydrolysis of the MBW anaerobic digestate was characterized by a surface tension (γ) of 49 mN/m at 1 g/L concentration in water compared to γ 61–63 mN/m at 2–3 g/L for the hydrolysates of three different MBW composts [10,11]. The product obtained by hydrolysis of the MBW anaerobic digestate was further processed by oxidation in plain water in the presence and absence of added hydrogen peroxide at 0.1–3 H_2_O_2_ moles per mole of organic C contained in the pristine digestate and in the presence or absence of irradiation with simulated solar light [17]. The measured surface tension of these products at 2 g/L in water ranged from 69 to 46 mN/m, with the lowest values being 46–49 mN/m measured for the products obtained in the presence of 0.25–0.5 H_2_O_2_/C mol/mol, with or without irradiation [17]. Better surface activity properties were exhibited by the soluble products isolated from the collected MBW slurry by simple centrifugation and by hydrolysis and ozonization of the pristine MBW slurry [16]. In these cases, the surface tension values at 2 g/L concentration in water were 47 mN/m for the product obtained by centrifugation, 42 mN/m for the product obtained by hydrolysis, and 37–39 mN/m for the product obtained by ozonization [16]. None of these products matched the properties of the product obtained by ozonization of the anaerobic digestate hydrolysate, for which the measured surface tension at 0.47 g/L concentration in water was 38 mN/m [12].

The development of high-performance biosurfactants is an integral part of a wider research framework aiming to develop biorefineries for the production of new value-added chemical specialties from renewable feedstock [18,19]. The production of biosurfactants from MBW, as described in the above cited work [10,11,12,13,14,16,17] has, in principle, several advantages arising from the nature of the material used as feedstock and from the fermentation and/or chemical processes adopted. First, MBW is the most available negative cost feedstock produced worldwide [16]. Secondly, MBW fermentation occurs by the native microbial population [20,21] and does not require the production and maintenance of selected microorganisms [22,23]. Thirdly, the hydrolysis and oxidation reactions belong to a well-assessed technology in the conventional chemical industry [23,24,25,26]. These features are most likely to attract the interest of major chemical companies in undertaking the implementation of MBW-based products at the industrial and commercial levels [27,28,29]. Yet, for this perspective to be realized, a number of criticalities need to be overcome.

Recently, soluble biopolymer (SBP) obtained by hydrolysis of the MBW anaerobic digestate has been reported to react with plain water to catalyze the production of O and OH radicals and undergo self-oxidation [17]. The reaction caused some depolymerization of the pristine SBP, production of organic carboxylic moieties, and partial mineralization of the SBP organic matter to CO_2_. The recovered oxidized polymer had a molecular weight in the range of 150–750 kDa, compared to the molecular weight > 750 kDa of the pristine SBP. Depolymerization was enhanced in the reaction performed under light irradiation [17]. In this case, the recovered oxidized polymer had a molecular weight in the 20–50 kDa range. For the recovered polymers, the measured surface tension values at 2 g/L concentration in water were in the range of 69–56 mN/m. Depolymerization and mineralization of the pristine SBP organic matter were found to increase for the reaction performed in the presence of hydrogen peroxide at 0.25–0.5 H_2_O_2_/C mol/mol. However, the low yields of the recovered oxidized polymers were offset by their improved surface activity properties; the measured surface tension for these products was 46–49 mN/m at 2 g/L concentration in water solution.

The above results [17] offered scope for investigating further the autocatalytic properties of SBP in order to develop a process that enabled better control of the rate of SBP chemical oxidation, reduced the depolymerization and mineralization degree, and improved the surface activity properties of the oxidized polymer. The present work reports the results obtained for the reaction of SBP with water containing hydrogen peroxide in catalytic amounts at H_2_O_2_/C µmol/mol level. Undoubtedly, obtaining valuable products by just keeping SBP in a water solution is the most desirable of low-cost processes, as it would not involve energy and reagent consumption and process waste needing secondary treatments for their disposal. To the authors knowledge, a similar oxidation process has never been reported.

## 2. Materials and Methods

### 2.1. Materials and Treatments

SBP was available from previous work [10,11]. It was obtained via hydrolysis at pH 13 and 60 °C from the anaerobic digestate of unsorted municipal food waste. The digestate was supplied by the Italian Acea Pinerolese MBW treatment plant located in Pinerolo (TO). The dry SBP hydrolysate was dissolved in plain water at pH 10 to obtain a solution containing SBP at a 1 g/L concentration. An aliquot of the solution was used as a control against other aliquots of the same solution irradiated with simulated solar light in the presence and absence of added hydrogen peroxide at 3 × 10^−5^ H_2_O_2_ moles per SBP carbon mole. After 5–6 days of irradiation, the irradiated solutions were acidified by the addition of 5 mL of 37% HCl per liter of solution. The acid solution was centrifuged at 12,000 rpm to separate the soluble and insoluble matter, which was lyophilized. The irradiation treatments were performed in a cylindrical photoreactor (height = 35 cm; diameter = 25 cm), in which four led lines of 78.5 cm were installed. The technical characteristics of the led irradiation systems (supplied by Universo S.R.L., Napoli, Italy) were as follows: SMD 2835, 120 LED/m, 10 W/m, 900 LM(lumen)/m, and light 4000K (natural light).

### 2.2. Products’ Isolation and Characterization

The dry crude soluble and insoluble products were re-dissolved in alkaline water at pH 9.7 at 1 g/L concentration and to a total volume of 0.25 L and then further processed by sequential vertical flow filtration using a lab-scale Amicon 8400 Stirred Cell (400 mL, Merck Millipore, Milano, Italy; Appendix A) operating under 3.5 bars N_2_ of pressure. Sequential filtration was carried out through different polysulphone membranes with decreasing molecular cut-offs to collect the retentates at 500 kDa (R500), 200 kDa (R200), 100 kDa (R100), 50 kDa (R50), 10 kDa (R10), and the final permeate at 0.2 kDa (P0.2). The obtained retentate and permeate fractions were dried at 60 °C to a constant weight and analyzed for their volatile solids, ash, C, and N content. The products were characterized for the C type and functional composition by ^13^C solid-state NMR spectroscopy and the surface tension in a water solution, as in previous work [12]. The critical micellar concentration (cmc) was calculated by measuring the surface tension at different concentrations of the added product in solution (0.01–4 g/L) [12].

### 2.3. Analytical Methods

^13^C solid-state NMR spectra were recorded, as previously reported [12,13,14,15,16,17], at 67.9 MHz on a JEOL GSE 270 spectrometer equipped with a Doty probe, available by JEOL (ITALIA) S.p.A. located in Basiglio (MI), Italy. The cross-polarization magic angle spinning (CPMAS) technique was employed, and for each spectrum, about 104 free induction decays were accumulated. The pulse repetition rate was set to 0.5 s and the contact time to 1 ms; the sweep width was 35 KHz, and MAS was performed at 5 kHz. Under these conditions, the NMR technique provides quantitative integration values in the different spectral regions [15]. Thus, the relative composition of C types and functional groups for each product in the text below is based on the integration of the band areas in the ^13^C NMR spectrum falling in the following chemical shift (δ, ppm) ranges: 0–53 for aliphatic (Af) C, 53–63 ppm for amine (NR) and methoxy (OMe) C, 63–95 ppm for alkoxy (OR) C, 95–110 ppm for anomeric (OCO) C, 110–160 ppm for total aromatic (Ph) C, and 160–185 ppm for carboxylic and amide (COX, X = OR, OM, NR, R = H, alkyl and/or aryl) C. The total integrated band area was assumed to represent the total C moles in the analyzed sample.

Potentiometric titrations were performed by Titrando 888 (Metrohm, Herisau, Swizerland) equipped with a Metrohm Unitrode combined glass electrode 6.0259.100 in the 2.0–11.5 pH range with 0.1 mol/L aqueous HCl. A total of 0.0200 g of the sample was dissolved in 30 mL of ultrapure water, and 5.0 mmol/L of KOH were added. Titrations were performed in a N_2_ atmosphere to avoid CO_2_ contamination by bubbling the gas directly into the solution. The electrode couple was standardized in terms of pH = −log[H^+^] by titrating a 5.0 mmol/L of KOH solution with HCl 0.1 mol/L.

The IR spectra were recorded, and band assignments were given according to the literature [13]. All other analytical and product characterization details were as previously reported [10,11,12,13,14,15,16,17].

## 3. Results

### 3.1. The Reaction of SBP and Water

Table 1 reports the experimental conditions adopted in the present work. The behavior of the SBP solution under irradiation with simulated solar light was investigated using a photoreactor specifically designed for this purpose (see Section 2). The trials with this reactor were carried out at 3 × 10^−5^ H_2_O_2_/C mol/mol ratio in order for hydrogen peroxide to perform, presumably as an initiator of the production of O and OH radicals, and to verify whether, under these conditions, mineralization and depolymerization of the organic matter might be reduced, while increasing the yield and improving the properties of the oxidized biosurfactant.

According to the experimental plan shown in Table 1, a preliminary trial (No.0) was carried out by irradiating the aqueous SBP solution at pH 9.6 for seven days. During this time, the solution pH decreased from the initial pH of 9.6 to 6.5. Four more trials (No.1–4) were carried out by irradiating the alkaline SBP solution in the presence of hydrogen peroxide at 3 × 10^−5^ H_2_O_2_ moles per mole of SBP carbon (C). In trial No.1, during five days of irradiation, the solution pH decreased from the initial pH 9.8 value to 6.5. In trial No.2, during the first seven days of irradiation, the solution pH was kept constant at 9.8 by adding KOH. In the following 12 irradiation days, no KOH was added, and the solution pH decreased to 7. Irradiation without the addition of KOH was continued for nine more days, and no change in pH occurred. Trials No.3 and No.4 were carried out under irradiation for six days, without and with the addition of KOH, respectively.

Figure 1 shows the trend of the solution pH versus time for the irradiation treatments of the SBP solution in the absence of added H_2_O_2_ (No.0) and in the presence of added hydrogen peroxide at 3 × 10^−5^ H_2_O_2_/C mol/mol (No.1 and No.3) without pH control. The decrease in pH in trial No.0 indicates the production of acid material during the seven days of irradiation in the absence of added hydrogen peroxide. By comparison, the pH of the not-irradiated SBP solution in water decreased to 9 after 14 days. This confirms the behavior of SBP to catalyze its self-oxidation by water, which has already been reported in the previous work [17]. The plots for trials No.1 and 3 indicate that the rate of production of acid material in the presence of hydrogen peroxide in the first 40 h is much higher than in trial No.0. Under longer irradiation times, the pH values recorded for the trials in the presence and absence of hydrogen peroxide seem to approach the same end value. In essence, the values for trials No.0 and No.3 seem significantly different. However, considering the values for trial No.1, it seems that the total final acid production was not statistically different in all of the trials. Overall, the experimental data are consistent with the role of hydrogen peroxide in acting as an initiator of the production of O and OH radicals and so to synergically support the autocatalytic action of SBP.

### 3.2. The Products’ Solubility and Molecular Weight

To obtain further insight into the nature of pristine SBP and of the oxidized products, the crude materials were fractionated based on their solubility and molecular size. The aqueous solutions of pristine SBP and of the products obtained in trials No.0, No.3, and No.4 were acidified at pH < 1, and the soluble and insoluble phases were separated by centrifugation. Table 2 reports the mass and C balance for the materials recovered from the acidification treatment of the SBP solutions obtained according to the experimental conditions reported in Table 1. From the available data for the soluble and insoluble materials, the sum of the soluble and insoluble mass and C yields range, respectively, from 95 to 101%, averaging 97.5 ± 2.6, and from 78 to 88%, averaging 82.3 ± 4.3. The data show that the recovered insoluble matter at pH < 1 decreases upon irradiation of the SBP solution and in the presence of hydrogen peroxide, whereas the soluble matter increases. A similar trend was observed in the ozonization of SBP [13]. In this case, the increased solubility in water at pH < 1 correlated with a decrease in the aromatic C in the oxidized product.

The crude soluble and insoluble matter in Table 2 was processed by sequential filtration through nine membranes, with the cutoff decreasing from 750 kDa to 0.2 kD. The process yielded retentates (Ri) and final permeates (Pi) with different molecular weights: i.e., ≥750 kDa for R750, ≥500 for R500, from 500 to 200 kDa for R200, from 750 to 150 kDa for R150, from 150 to 100 kDa for R100, from 200 to 50 kDa for R50, from 50 to 10 kDa for R10, ≤10 kDa for P10, from 20 to 5 kDa for R5, ≤0.2 kDa for P0.2. Table 3 reports the process mass yields of the fractions isolated by membrane ultrafiltration relative to the not-irradiated crude pristine SBP. For each material listed in Table 3, the sum of the insoluble and soluble fractions’ mass ranges from 99 to 107%, averaging 103 ± 4. It may be observed that the R750 fraction is the major fraction, accounting for 58% of the total mass recovered from the ultrafiltration of not-irradiated pristine SBP. By comparison, for all insoluble and soluble crude products obtained from the irradiated SBP solution (treatments No.0, No.3, and No.4), the highest molecular weight R500 fraction accounts for only 2–9% of the total recovered mass, while the major fractions are R200, R10, and P10, each accounting for 22–37% of the total recovered mass. The data indicate that irradiation of SBP causes depolymerization of the pristine SBP. The insoluble matter recovered from all irradiated SBP solutions (No.0, No.3, and No.4) is characterized by a much higher content of R200 fractions compared to the P10 fractions, while the contrary is true for the soluble matter. The depolymerization of the irradiated SBP in the absence of added hydrogen peroxide, together with the observed pH decrease (Figure 1), is a further argument supporting the autocatalyzed reaction of SBP with water. Compared to treatment No.0, the addition of hydrogen peroxide to the irradiated SBP solution (treatments No.3 and No.4) increases the relative yield of the lower molecular weight R10 and P10 soluble fractions, although the recovered insoluble crude products have a higher R200 high-molecular-weight fraction content.

Considering the complexity of the chemical composition and the different solubilities and different plausible interactions with the different membranes, it may be concluded that both the applied acid precipitation and ultrafiltration methods allow a reliable and satisfactory recovery of the total mass of the treated raw products.

### 3.3. The Products’ Surfactant Properties

The surface tension of the fractions with a molecular weight ≥ 100 obtained in treatments No. 0, 3, and 4 was measured in order to assess their surface activity properties in comparison with the fractions obtained from the not-irradiated crude pristine SBP and from the ozonized SBP. Measured indicators of product quality were the surface tension at the product 2 g/L and at the critical micellar concentration (cmc), and the product color in the aqueous solution. For SBP biosurfactants, the product’s black color was found to be a drawback, limiting its use because of the active principle in formulations applied as fabric detergents or auxiliaries for fabric dyeing [12,17].

Table 4 shows that the water solution of all fractions isolated from the crude products obtained in trials No.0, No.3, and No.4 has a remarkably lower surface tension compared to the fractions composing the not-irradiated pristine SBP. What is particularly relevant are the R200 fractions obtained in treatments No.0 and No.4, which are characterized by the lowest cmc and γ_cmc_ values matching the values reported for the R150 fraction isolated from the crude pristine SBP ozonized for 64 h [12].

Figure 2 shows that that the color darkness of the products in the 2 g/L solutions decreases in the order No.0 R200 > No.3 R200 > No.0 R100 = No.4 R200 > No.4 R100 > No.3 R100.

### 3.4. The Relationship between the Products’ Chemical Composition and Surfactant Properties

#### 3.4.1. Products’ Characterization by ^13^C Solid-State NMR Spectroscopy

The products in Table 3 and Table 4 were analyzed by solid-state ^13^C NMR spectroscopy for the content of the C type and functional groups in order to find a relationship between their surface activity properties, color, and chemical composition. Figure 3 and Appendix A report the relative C mol/mol % composition of the C types and functional groups, relative to the total organic C, for the major fractions with a molecular weight ≥ 100 obtained from the membrane ultrafiltration of the not-irradiated crude pristine SBP and of the insoluble matter of the irradiated SBP solution listed in Table 3.

The relative C mol/mol % composition for each product is based on the integration of the band areas in the 13 C NMR spectrum arising from the resonance of the different C types and functional groups, measured in the following chemical shift (δ, ppm) ranges: 0–53 for aliphatic (Af) C; 53–63 ppm for amine (NR) and methoxy (OMe) C; 63–95 ppm for alkoxy (OR) C; 95–110 ppm for anomeric (OCO) C; 110–160 ppm for total aromatic (Ph and PhOY; Y = H, R, Ar) C; and 160–185 ppm for carboxylic and amide (COX, X = OR, OK, NR, R = H, alkyl and/or aryl) C. The total integrated band area was assumed to represent the total C moles in the analyzed sample (see Section 2).

The data in Figure 3 show that the not-irradiated and irradiated SBP have a very complex composition, made by macromolecular pools with different molecular weights and chemical compositions. This feature is inherited from the pristine MBW containing the natural lipids, polysaccharides, proteins, cellulose, and lignin proximates present in food waste [16]. It makes it very difficult to assess a reliable relationship between the chemical composition and surfactant properties of the crude products obtained from the treatments listed in Table 1. Yet, such a relationship is necessary for the implementation of MBW-derived biosurfactants at both the industrial and commercial scales.

Some useful information may be obtained from the analysis of the data in Figure 3. For example, No.0-R100 is distinguished for the highest content of aliphatic (Af) C and the lowest content of total aromatic (Ph + PhOY, Y = H, R, Ar) C compared to all other fractions. The total aliphatic (Af) and carboxyl (COX, X = O^−^, N) accounts for 75% of the product total C. (Af + COX)/(Ph + PhOY) = 12.6 is definitely the highest value compared to all other samples (see Figure 4). This feature suggests that this biopolymer may be the best for the manufacture of melt-extruded composite mulch films [12]. Yet, as a biosurfactant, No.0-R100 does not rank first.

For the surface tension and cmc values (Table 4), No.0-R200 and No.4-R200 rank first. They have a very similar chemical composition, if you compare one to the other. Their (Af + COX)/(Ph + PhOY) values are 3.9 for No.0-R200 and 4.4 for No.4-R200, being higher than 2.6 for the not-irradiated SBP but significantly lower than all of the other products.

For the solution color (Figure 2), the ranking order of decreasing darkness No.0 R200 > No.3 R200 > No.0 R100) = No.4 R200 > No.4 R100 > No.3 R100 is somewhat related to the respective values 3.6, 3.3, 4.3, 3.4, 2.0, and 1.2 for the products’ Af/COX C ratio (see Figure 5).

#### 3.4.2. The Products Hydrophilic Lipophilic Balance (HLB)

Hydrophilic lipophilic balance (HLB) is used to optimize the performance of commercial products containing surfactants as active principles. For example, emulsifiers are classified on the basis of their HLB value [30]. This indicator is calculated based on the relative amounts of hydrophilic and lipophilic C types and functional groups to which a specific group contribution value is attributed [31]. The calculated HLB is reported for single surfactants with a definite known chemical composition, e.g., sodium oleate, polyethylene glycols, and sorbitans [30]. Other authors [32] have questioned the methods and results obtained in the determination of HLB. They suggest that published HLB should only be taken as approximate guidelines.

In the case of MBW-derived biosurfactants containing a mixture of different molecules, the probability of understanding the relationship between their chemical composition and behavior in water solution is rather low. The behavior of each molecule may be affected by interactions with other different molecules. Yet, in the case of ozonized [12] and hydrogenated [33] SBP, a high correlation was established according to the equation.
γ = a HLB + b(1)

In Equation (1), γ is the surface tension value measured at a 2 g/L concentration in water for several products with different molecular weights in the 30–750 kDa range obtained at different reaction times, and HLB is calculated (see Appendix A) from the content of C type and functional groups obtained by ^13^C spectroscopy (Figure 3) and from the specific group contribution number (gn) for each C type and functional groups reported in the specialized literature [31,34,35]. Figure 6 shows the γ vs. HLB plot for all fractions (Ri) obtained by membrane ultrafiltration of crude products obtained in trials No.0, No.3, and No.4 listed in Table 3 for the not-irradiated crude pristine SBP and the 10 fractions with molecular weights in the 30–750 kDa range obtained by ozonization of pristine SBP at different reaction times [12].

For the scope of the present work, the irradiated catalyzed No. 0-R200 and No. 4-R200 samples were of primary importance due to the fact that they exhibited the lowest remarkable surface tension (34–35 mN/m), close to that of the ozonized SBP R150 and R750 fractions (37–39 mN/m). The Grubbs’ test was applied to γ values to evaluate if these samples could be considered outliers. The results show that, at a significance level of 0.05, the samples cannot be discarded from the population. In order to highlight the correlation between γ and HLB, the Person correlation coefficient was calculated. The parameters show a significant correlation at the 0.05 level with a correlation coefficient of 0.62 (*p*-value = 0.008). However, the γ-HLB correlation is consistent with the recommendation of Pasquali et al. [32] to use the HLB parameter as an approximate guideline. Indeed, the group contribution values [31] used for the calculation of HLB (Appendix A) do not account for the change in polarity of a specific C type bonded to different functional groups. For example, in poly-functionalized C chains, the different functional groups may modify the hydrophilic or lipophilic properties of the product. Furthermore, for the MBW-sourced biosurfactants dealt with in the present work, the variability of the pristine MBW may affect the reproducibility of the biosurfactant obtained with the same production process, but from different production batches. The major contributors to the higher HLB values were the NR and COX functional groups, to which the respective groups contribution number 9.4 and 2.2 were assigned. These numbers, taken from the literature [31,35], were tentatively used to calculate the HLB values for all products corresponding to the data points in Figure 6. The two NR and COX functional groups accounted for about 65% of the total HLB value of the product (Appendix A). The literature reports several different group contribution number values for the above NR and COX functional groups, depending on the structural unit to which they belong (Table 5).

The data in Table 5 do not help understand why the water solutions of the two No.0-R200 and No.4-R200 irradiated samples and the R150 fraction obtained from the pristine SBP ozonized for 64 h (Table 4) exhibit the same surface tension values and bleached color, although the products have quite different chemical compositions. The large differences in the group contribution numbers reported in Table 5 show how different carboxyl functional groups can affect the HLB and, in turn, the surface activity properties of biosurfactants. Indeed, the performance of SBPs in their diversified applications has been found to be connected to their acidic sites [11].

The functional groups of different acid strengths and of electrons donating and accepting power present in SBPs have been found to be capable of affecting their solubilities in water at different acid pHs and affecting the products performance for their use in different sectors of the chemical industry, agriculture, environmental remediation of polluted soil, and waste waters. As ^13^C NMR spectroscopy does not allow the speciation of the COX and PhOY functional groups reported in Figure 3, further investigation was carried out in the present work by potentiometric titration and IR spectroscopy to obtain the breakdown of COX and PhOY into carboxylate (-COO^−^), amide (-CONR_2_), phenol (PhOH), and phenoxide (PhOR) groups.

#### 3.4.3. Determination of Carboxylate (-COO^−^), Amide (-CONR_2_), Phenol (PhOH), and Phenoxide (PhOR) in the Products Obtained by Irradiation of SBP

Potentiometric titrations were performed in the 2.0–11.5 pH range with 0.1 mol/L aqueous HCl. The analyzed sample (0.02 g) was dissolved in 30 mL of ultrapure water containing 5.0 mmol/L of added KOH. Figure 7 shows a typical titration curve with the corresponding first derivative function and the two equivalent points (EP1 and EP2) that can be detected for all the samples. As described in Section 2, the plot allows the content (Table 6) of the three types of acid functional groups with different acid strengths to be measured, i.e., PhOH and the -COOH I and -COOH II groups belonging to different substituted aliphatic and aromatic carboxylic moieties.

The data in Table 6 were used to obtain the breakdown of the COX and PhOY C concentrations determined by ^13^C spectroscopy (Figure 3) into carboxylate (-COO^−^), amide (-CONR_2_), phenol (PhOH), and phenoxide (PhOR) groups, which are reported in Table 7.

IR spectroscopy measurements were carried out to support the results of the titration measurements. The IR spectra (Appendix A) for the products obtained from SBP (Table 7) were qualitatively similar to those registered [13] for the ozonized SBPs obtained from urban green (CV) and green food waste (CVD) composts. As for the SBPs, for all SBP products, the IR spectra (Appendix A) confirmed the presence of all functional groups listed in Figure 3. The spectra show broad bands above 2000 cm^−1^ arising from H-bonded O-H and N-functional groups, sharper bands at 2800–3000 cm^−1^ arising from C-H stretching vibrations superimposed over the broad O-H and N-H bands, strong bands with peak absorption at 1643 and 1557 cm^−1^ arising from carboxylate (COO^−^) and amide (CON) functional groups, respectively, bands at 1391 cm^−1^ arising from C-H bending vibrations, and bands at 1240 and 1033 cm^−1^ arising from glycosidic linkage vibrations.

The IR absorption spectra (Appendix A) are consistent with the presence of carboxylate and amide functional groups. The data in Table 7 show that the COO^-^ functional groups account for 32–72% of the total COX content reported in Figure 3. It is evident how the large variabilities of the -COO^−^ and -CON relative contents in the products and of the groups numbers assignable to these groups (Table 5) can greatly affect the calculated HLB and the tentative correlation reported in Figure 6. Unfortunately, data for the COOH and PhOH content for the ozonized SBP samples in Figure 6 are not available. This makes it impossible to recalculate HLB values for the ozonized samples in Figure 6 and assess the magnitude of the effect of the new values on the correlation with the surface tension of the 17 experimental points.

## 4. Discussion

### 4.1. Key Findings in the Present Work

The results reported in Section 3 prove that it is possible to overcome the drawbacks shown in the previous work [12,13,14,16,17] through a process exploiting the autocatalytic properties of SBP. The autocatalysed process allows controlling better the rate of the SBP chemical oxidation, reducing the depolymerization and mineralization degree, and improving the surface activity properties of the oxidized polymer. These findings confirm previous knowledge that, although oxidation of organic carbon to CO_2_ is the most thermodynamically favored reaction, the mineralization of organic molecules is markedly slower than their de-aromatization, the latter involving ring opening with the formation of aliphatic carboxylic and hydroxylated C moieties [12,13,14,16,17,33] whose mineralization is the slowest step [36].

Performance wise, the irradiated SBP samples obtained under the experimental conditions reported in the present work are competitive biosurfactants with the ozonized SBP samples, but they are much lower in cost.

The data in Figure 1 prove that a very small amount of hydrogen peroxide present in water can act as an initiator of the production of O and OH radicals and so can synergically support the autocatalytic action of SBP. The data show that the decrease in the pH at the end of the reaction is the same in the absence and in the presence of H_2_O_2_ and cannot be justified on the basis of a stoichiometric oxidation reaction between SBP C and H_2_O_2_ since the added H_2_O_2_/C ratio in the reaction medium was 3 × 10^−5^ mol/mol. Under these circumstances, the decrease in pH can only be explained by a reaction between SBP C and water performing as a solvent and ultimate oxidant. To the authors’ knowledge, this is a unique example of the use of H_2_O_2_ as a co-catalyst in the presence of H_2_O as the terminal oxidant for the oxidation of an MBW-derived material (SBP) performing as substrate and catalyst.

From a practical perspective, the experimental conditions adopted in the present work (Table 1) were an attempt to reduce the depolymerization of the substrate and enhance the yield and quality of the high-molecular-weight biosurfactants obtained by oxidation of the SBP, with ozone and hydrogen peroxide as terminal oxidants. Table 3 shows that the irradiation of substrates in the absence and presence of H_2_O_2_ caused some depolymerization of the pristine SBP, but the total mass yield of the recovered fractions with a molecular weight above 100 kDa ranged from 45 to 59% and did not decrease for the treatments in the presence of hydrogen peroxide at the added H_2_O_2_/C 3 × 10^−5^ mol/mol ratio. Table 4 shows that all of the fractions that were insoluble at acid pH with a molecular weight above 100 kDa yield aqueous solutions with a surface tension ≤ 48 m N m^−1^ at 2 g/L and cmc ≤0.5. These results are quite remarkable considering that neither the oxidation of SBP in the presence of hydrogen peroxide at 0.1–3.0 H_2_O_2_/C mol/mol [17] nor the ozonization of MBW [16] and SBP [12] achieved the same yield of high-molecular-weight biosurfactants, coupled with the remarkable surface activity properties, as obtained in the present work. It is quite evident that, under these circumstances, the reported oxidation process is the most sustainable chemical process, conceivable from environmental and economic viewpoints.

### 4.2. Process Scale-Up from Proof-of-Concept to Industrial Level

In the present work, the photo-oxidation of SBP was performed in a cylindrical photo reactor operated in batch mode for 5–7 days to obtain the crude oxidized product, which was then further treated according to the following scheme 1 in Figure 8 (also see Section 2).

It may be observed that, while the photo-oxidation step required no reagent consumption, except water, the secondary treatment of the oxidized crude product required the consumption of HCl and NaOH. In this fashion, the best biosurfactants (No.0-R200 and No.4-R200 in Table 3) were obtained in 24 and 37 mol/mol % C yield, relative to the pristine SBP C. The yield and valuable properties of the biosurfactants offer worthwhile scope for further work to upgrade the sustainability of the process in scheme 1 by reducing the consumption of the acid and alkali reagents. This can be feasibly achieved through a continuous plant comprising a tubular reactor, where the SBP residence time was optimized to obtain the highest product yields and the added mineral acid and alkali were recovered and recycled to the plant section dedicated to the crude product secondary treatment, according to scheme 2 in Figure 9.

So far, most photoreactors have been designed for use in the decontamination of industrial waste water [36] in order to optimize, case-by-case, the mineralization of the contaminants into carbon dioxide, water, and inorganics. To this end, many oxidation processes are reported in the literature, such as TiO_2_/UV, H_2_O_2_/UV, photo-Fenton, and ozone (O_3_, O_3_/UV, and O_3_/H_2_O_2_). Their main disadvantage is their high cost. UV radiation generation by lamps or ozone production is expensive. These processes could be improved through the use of catalysis (photo-Fenton and heterogeneous catalysis with UV/TiO_2_) and solar energy. The natural sunlight-driven photoreactor depicted in scheme 2 has been used by the authors of the University of Torino and co-workers from other institutions [37] to validate the results obtained at lab level in a closed Pyrex cell with a Xenon (1500W) lamp (Solarbox) equipped with a 340 nm cut off filter for the decontamination of waste water catalyzed by compost-derived SBPs as photosensitizers. A similar continuous tubular reactor, specifically designed to optimize the first photo-oxidation step of scheme 2, might reduce the irradiation time to make the same or better-performing biosurfactants obtained in the batch reactor depicted in scheme 1.

Industrial large-scale solar photocatalytic plants are known for operating the production of solar fuel from solar energy and water and for the synthesis of bulk commodities such as solvents and fuels, as well as chemicals for niche applications [38,39,40,41,42,43,44,45]. The application of photocatalytic technology in water for processing biomass as feedstock yielding solar value-added chemicals [46] is particularly relevant in relation to the perspectives of scaling up the proof-of-concept work depicted in scheme 1 to the industrial level by scheme 2. The oxidation of biomass and its derivatives has been reported to be capable of yielding many value-added chemicals, such as gluconic acid [47,48], levulinic acid [49], lactic acid [49], arabinose [50], and formic acid [51], which are precursors of pharmaceuticals, food additives, cosmetics, and polymers. These reactions have been carried out using TiO_2_ coupled with other oxides, WO_3_, CdS/CdOx, zeolite, heteropolyacids (HPAs), and carbon nitride, with the intent of depolymerize the polymeric lignocellulose biomass proximates and to obtain simple small molecules to use for the manufacture of finished consumers’ products and hydrogen production. For these reactions, it has been found [46] that the main drawback is the low selectivity of the radical formed oxidizing species, which indiscriminately attack both the starting substrate and the formed intermediates. This presents scope for further research focused on understanding the reaction mechanism and adopting strategies aimed at tuning appropriate physicochemical parameters of the catalysts, the reactor configuration, the solvent, and the pH in order to improve the efficiency of the process.

The low reaction selectivity reported for the manufacture of the above chemicals [47,48,49,50,51,52] has also been challenged in the oxidation of MBW [16] and the SBP derivatives [10,11,12,13,14,17], aiming to reduce the depolymerization and mineralization of the organic matter of the pristine substrates and obtain polymeric biosurfactants that are ready for use. The process depicted in scheme 1 has succeeded in significantly improving reaction selectivity. Yet, the process scale according to scheme 2 will have to account for the same factors evidenced for the photocatalytic processes aiming to depolymerize the organic matter of the pristine biomass and obtain the simple molecules for use as building blocks for the manufacture of ready-for-use products [46]. Given that the productivity of photo-chemical reactors scales with the surface area rather than the volume of the reactor, for the scale-up of the process demonstrated in the present work to an industrial level, the issue is using convenient natural light source or commercial light sources and solar simulators, both of which have their own limitations. The economic viability of these options needs to be assessed case by case, depending on the geographical sites, local climate conditions, and the type of reaction and photoredox catalysts. Thus, reactor design must account for all these factors.

In the case of the present work, moving from basic research to applied technology will take a lot of concerted work by chemists and chemical engineers. To appreciate the novelty and the potential impact of the authors results reported here, it should be acknowledged that, contrary to the objectives of the present work, so far, research work on photo-chemical processes has been focused on enhancing the depolymerization of the biomass natural proximates to obtain small molecules [36], referred to as building blocks, to use for the synthesis of consumers finished products and materials. Due to the relatively low market value of these products, this strategy is less attractive in terms of economic feasibility than the production of biofuel. The situation is very different in the case of the biosurfactants obtained in the present work, which may reach potential market values of one to two orders of magnitude higher than the above building blocks.

### 4.3. Improving Biosurfactants’ Properties

Based on the global market size of surfactants [1,2,3] and the prospected growth [5,6,7,8] of the biosurfactants’ market (see Section 1), one can readily deduce that the quality and performance of the biosurfactants obtained in the present work are key factors for novel biorefineries [18,19], as, for example, a refinery using MBW as renewable feedstock [12,13,14,16,17] to be competitive with oil-based refineries. To this end, bacterial sophorolipids and rhamonlipids biosurfactants [6] represent model products to be matched for performance. These biosurfactants have glycolipids with a chemical structure consisting of either one or two rhamnose molecules attached to one or two β-hydroxydecanoic acids [53,54]. With an average mass ranging from 500 to 1500 Da, biosurfactants are widely divided into low-molecular-mass and high-molecular-mass biosurfactants [53]. They lower the surface tension of water from 70 down to 28 mN/m at the critical micellar concentration of 0.8–2 g/L [55,56]. Their market price ranges from 30 to 150 EUR/kg [7], which is much higher than the 1.5–5.5 EUR/kg price [5] of fossil sourced from synthetic or oleochemical natural surfactants [53,54].

The SBP Ri fractions listed in Figure 3 contain all of the C types present in the above bacterial biosurfactants [55,56,57] together with aromatic C. In the Ri fractions, these C types represent the memories of the lipids and lignocellulosic proximates of the pristine materials chain from which they are derived, i.e., the crude as collected MBW, the crude SBP, and the oxidized SBP. For the products reported in Table 4, which were analyzed for the surface activities, Appendix A reports the C types and functional groups composition and the values of several empirical parameters based on the sum and/or ratios of the C types and functional groups mol/mol % values. The values of the empirical parameters were analyzed against the γ (2 g/L) values given in Table 4. The best correlation with the product surface tension values was obtained for the values of the total C content as a sum of the Af, OR, OCO, and COX carbon types. Figure 10 reports the plot of the total Af, OR, OCO, and COX C versus the product surface tension values. The correlation coefficient (R) calculated over all nine experimental points result is 0.19. Exclusion of the experimental points for No.0-R200 and No.4-R200 yields the linear regression equation with R = 0.93
γ = 0.63 (Af + OR + OCO + COX) + 95(2)

A better correlation, including the No.0-R200 and No.4-R200 products, is obtained using the COO^−^ values in Table 7 in place of the COX values. Figure 11 shows the plot of the surface tension against the total (Af + OR + OCO + COO^−^)/COO^−^ values for the not-irradiated SBP and the retentate fractions (Ri, i = 200 and 100). The data fit the linear equation
γ = a (Af + OR + OCO + COO^−^)/COO^−^ + b(3)
for which a = −2.20, the intercept b = 65, and the correlation coefficient R = 0.89. The results of the plot in Figure 11 are consistent with the plot in Figure 12, which includes the experimental points of the ozonized samples of two different SBPs obtained in previous work [13,14] from composts of urban green gardening residues (CV) and of mixed green residues and unsorted food waste anaerobic digestate (CVD). For Equation (3), interpolating the 20 experimental points in Figure 12, a = −2.36 ± 0.38, b = 70.4 ± 2.9, R = 0.82. The 0.82 correlation coefficient is quite remarkable considering that the data in Figure 12 and Appendix A relate to different pristine materials processed under different experimental conditions and at different dates and to products with different molecular weights and compositions and were obtained using different analytical methods and assumptions for data elaborations. The data show that the irradiated SBP solutions exhibit lower surface tension than all of the other samples.

For the present discussion, the empirical (Af + OR + OCO + COO^−^) parameter containing the hydrophilic carboxylate concentration values in Table 7 is assumed to better represent the virtual total content of rhamnolipids-like organic moieties compared to (Af + OR + OCO + COX) parameters. Both Figure 11 and Figure 12 confirm that increasing the relative content of the virtual rhamnolipids-like organic moieties, represented by the total content of the liphophilic Af, OR, OCO, and hydrophilic COO^−^ C moles per mole of hydrophylic COO^−^ carboxylate functional groups, can significantly improve the surface activity of the irradiated SBP products. This finding sets the specific target composition to be achieved for the product to be obtained through the development of the irradiation process according to scheme 2.

### 4.4. Challenges and Perspectives for Improving Processes and Products Sustainability: A Comparative Review of Chemical and Bacterial Surfactants

Feedstock and productivity are common issues for all biobased products that need to improve their production processes. Their impacts on the overall product sustainability depend on the type of technology used for the product manufacture. For example, in the case of the chemical surfactants listed in Table 4, obtained by the chemical-photochemical technology depicted in schemes 1 and 2, the relative impacts of the costs of the feedstock and of the product manufacture process are much different from the case of the rhamnolipids and sophorolipids bacterial surfactants produced exclusively through biotechnology. Rhamnolipids and sophorolipids are referred to as second-generation biosurfactants [53,54,55,56,57], which are produced through biological processes (biocatalysis or fermentation). Compared to chemical processes, biological processes present severe cost drawbacks for their scale up to the industrial level. Synthetic surfactants are substantially less expensive than biosurfactants when comparing pricing: 0.8–2.2 USD/kg for petro-based; 1.3–2.6 USD/kg for oleochemical; and 20–40 USD/kg for rhamnolipids and sophorolipids. In addition to the high price, a major drawback for the implementation of biosurfactants used as feedstock biomass obtained from dedicated cultivations is societal disagreement over the true cradle-to-gate effects of using land for the production of renewable chemical feedstock rather than for food production. This is not the case for MBW-sourced biosurfactants produced by photochemical oxidation, according to schemes 1 and 2.

So far, the R&D work on rhamnolipids and sophorolipids has attempted to improve their biosynthesis [53,54,55,56], while chemical technology has been mostly applied for the isolation and purification of the crude product obtained by fermentation. Chemical oxidation with O_3_ and H_2_O_2_ in water, combined with genetic engineering, has been applied to produce sophorolipids’ derivatives [58] in order to expand the application areas of the bacterial surfactants and establish an efficient, scalable technological basis for their competitive production at the industrial level. Still, prior to ozonolysis, most of the R&D work has been focused on genetic engineering in an attempt to reduce and simplify the multistep process for the production of the substrate to be ozonized, increase its efficiency, and reduce the end product cost. The investigated combination of genetic engineering and chemical oxidation did not assess definite exploitable benefits, which enabled a competitive industrialization of the process and products.

#### 4.4.1. Biowastes as Feedstock for Enhanced Products Sustainability

Enhanced sustainability of rhamnolipid and sophorolipid biosurfactant production has been attempted by employing second-generation feedstocks [59,60,61,62], specifically industrial residual biomass waste streams such as sunflower oil, waste fried oil, jatropha oil, and animal fat (as hydrophobic substrates), and sugarcane molasses, soy molasses, glycerol (as hydrophilic substrates), and food waste. In this scenario, municipal unsorted food waste (MBW) and its anaerobic digestates are the most sustainable waste materials to use as potential waste feedstock. Their global production [61,62] is 1.3 billion/year. They are readily available in confined spaces, thanks to separate source collection practices practiced by municipalities [63]. As collection costs are paid off by citizens’ taxes, MBW is defined as a negative cost feedstock [64]. Together with urban settlements, farms are major producers of biodegradable waste comprising crop scraps, exhausted plant residues, and animal dejections, while conventional aerobic fermentation by native unsorted microbial populations is the most widely used waste treatment practice in Europe, producing, for example, 80 Mt/yr. of anaerobic digestate [65,66,67,68,69]. The widely abundant availability and low cost of MBW anaerobic digestates have spurred a lot of workers to use these materials as feedstock for the production of rhamnolipids and sophorolipids bacterial surfactants. Patria et al. [70] published the results of a techno-economic analysis of a fed-batch fermentation process utilizing food waste as a substrate for the production of rhamnolipids. The process comprised two fermentation steps: one for the production of the food waste digestate and the other for the production of the rhamnolipids from the MBW digestate, which operated simultaneously. The fermentation broth was centrifuged, filtered, extracted, and purified from the cell-free supernatant through hexane extraction. The production cost of this process was estimated to be USD 37.5 per kg of 50% purity rhamnolipid syrup, against a selling price of USD 225/kg. The raw material cost for this process accounted for 10% of the total production cost, compared to 50% of the total production cost upon using hydrophobic substrates like vegetable oils. Relevantly, in this context, the production cost of the SBP obtained by chemical hydrolysis of the MBW anaerobic digestate, which was used as a feedstock for the production of the biosurfactants listed in Table 4 according to scheme 1, was reported to be 0.04 EUR/kg of the SBP 10% water solution, as assessed in the real operational environment of three waste MBW treatment plants in three different countries [71].

#### 4.4.2. Improving Biosurfactant Productivity

For the bacterial sophorolipids and rhamonlipids biosurfactants [6] representing model products to be matched for performance by the chemical surfactants according to scheme 2, the employed approaches to improve their productivity are the optimization of culture conditions, the selection of robust microorganisms, the genetic modification of microorganisms, and the development of novel cost-effective downstream processes [60,61,72,73,74]. In spite of the great amount of research, according to the authors of the present work, the major factor hindering the scale-up of the results obtained by previous workers to the industrial level is the application of processes based on biotechnology that do not allow a real, consistent breakthrough in the end product cost reduction. Biowastes used as secondary feedstock may contain some ingredients that inhibit the fermentation processes, resulting in limited biomass and low production yields. Although many different wild-type bacteria have been reported to produce the bacterial surfactants, none of the organisms that produce them obtain high enough yields for the products to be commercially viable [72,73]. Therefore, as with many biosurfactants, rhamnolipids fail to reach the marketplace and substitute for conventional chemical surfactants. In conclusion, bacterial surfactants must face several challenges in process design, downstream processing, and low yields [73], process adaptability and optimization, socio-economic impacts and sustainability benefits, and regulatory barriers [60,61,70]. Based on the results of the present work, the authors think that, rather than trying to improve rhamnolipids production by biotechnological process, the optimization of the photochemical process and rhamnolipid-like products, according to scheme 2, offers more perspective concerning the development of low-cost biosurfactants matching the properties and performance of rhamnolipids and sophorolipids, and therefore being competitive with current commercial surfactants derived from fossil sources from all viewpoints.

The authors’ previous work [16] addressed the issue of chemical technologies and biotechnologies as a means to valorize MBW as feedstock for the production of value-added products. Among the criticalities pointed out in the above sections for the production of the bacterial surfactants [60,61,72,73,74], another relevant drawback hindering the growth of the biobased chemical industry is the reluctance of major chemical companies to undertake biochemical technology as a core business [22,23,31,74]. Major chemical companies, e.g., Dow and BASF [53], are key actors that can play a major role in scaling R&D work to the industrial and commercial level. They are developing competitively friendly green processes and applying traditionally assessed, mature, and more familiar chemical technology to renewable feedstock rather than putting a lot of resources into dealing with the drawbacks of biotechnology.

Organic chemistry offers a wide range of chemical routes for converting monomers and building blocks into a variety of molecules for the chemical industry to manufacture bulk and fine chemicals and finished consumer’s products starting from fossil sourced hydrocarbons, e.g., hydrolysis, oxidation, alkylation, etherification, esterification, condensation, and polymerization. In principle, building on the current knowledge, renewable resources and bio-based feedstock may present a sustainable alternative to petrochemical sources to satisfy modern society’s ever-increasing demand for energy and chemicals. However, the conversion processes needed for future bio-refineries will likely differ from those currently used in the petrochemical industry and require the development of dedicated technology due to the high chemical complexity, entropy, and site-specific variability of biomass compared to fossil sources.

In the case of the production of SBP by chemical hydrolysis [71], used in the present work as feedstock for the production of the chemical biosurfactants according to scheme 1, a highly significant linear relationship was found between the chemical composition of the pristine MBW materials collected in different seasonal climate conditions and citizens’ food consumption habits of Italy, Greece, and Cyprus, chosen as case studies, and the composition of the SBPs. Yet, the induced variability in the SBP composition did not critically affect their performance as catalysts in a new MBW anaerobic fermentation process for the production of biogas and digestate with reduced ammonia content, which was replicated in the real operational conditions of the three countries.

In the case of the present work, photo-chemical oxidation (scheme 1) was an approach to improving the products’ surfactant properties. Chemical oxidation is a strategic reaction in the chemical industry to obtain added-value chemical intermediates from fossil oils and coal that are poor in oxygen. Green chemistry [75] is strongly influencing all aspects of chemical research and oxidation reactions [76]. In accordance with the twelve principles of green chemistry [77], oxidation reactions should evolve toward the exclusive employment of more benign, environmentally acceptable oxidants, such as O_2_ and H_2_O_2_. In this context, photocatalytic oxidation is attracting researchers’ attention as it allows one to use atmospheric oxygen directly as a terminal oxidant, essentially rendering the whole process “green” and more cost-efficient [37,45].

The oxidation reactions described in the present work (schemes 1 and 2) extend beyond these requirements since they employ water as the greenest solvent and terminal oxidant, together with waste-derived SBP as substrates and catalysts, and do not require the consumption of energy and reagents and the synthesis, recovery, and/or regeneration of expensive catalysts. Thus, the implementation of this system on an industrial level would constitute an example of the most desirable, eco-friendly, and cheapest process. Compared to the current catalytic systems used for photoredox transformations [36], the envisioned photo-oxidation system to be implemented according to scheme 2 does not require expensive ruthenium- and iridium-based photocatalysts or other relatively less expensive photoredox-active compounds containing no precious metals and heterogeneous semiconductors, such as titanium oxide, carbon-based catalysts, and 2D nanomaterials to promote water splitting and oxidative treatments. The simplicity of the proposed reaction system in schemes 1 and 2 may overcome the current challenges connected with the design and optimization of photo-reactors that are hindering the implementation of solar photo catalysis on the industrial and commercial levels. Reaching industrially relevant productivity and quality of biosurfactants through scheme 2 is likely to attract the interest of major chemical companies and MWB management companies to collaborate and invest resources in developing a new MBW-based plant integrating chemical and biochemical technologies. In this plant, the biochemical section produces the MBW digestate by conventional fermentation, not requiring selected or genetically modified microorganisms. The chemical section produces the SBP and the derived high-performance biosurfactants by conventional chemical hydrolysis of the digestate in water, followed by photo-oxidation in the absence of added oxidants. Under these conditions, the biosurfactant production cost should not be much different from the SBP production cost, estimated to be 0.04 EUR/kg of SBP 10% water solution [71], well below USD 37.5 per kg of 50% purity rhamnolipid syrup [70]. The perspective to operate the new MBW-based plant through conventional biochemical and chemical low-cost integrated technologies should spur the establishment of joint ventures between MBW management companies and chemical companies, with the required competencies in collecting and processing the MBW feedstock by fermentation to produce the anaerobic digestate, in the chemical hydrolysis and oxidation reactions to produce the SBP and derived biosurfactants, and in producing and commercializing the finished biosurfactants-based formulates for consumers’ use.

## 5. Conclusions

The experimental work reported in the present work provides proof-of-concept of a most sustainable chemical oxidation process for the production of value-added chemical specialties from renewable feedstock. The process has been demonstrated in the case study of municipal biowaste, which is the most sustainable negative-cost feedstock available. The applied integrated biochemical–chemical process comprises conventional MBW fermentation by the native microbial population and conventional chemical hydrolysis and photochemical oxidation of the MBW anaerobic digestate in water without added oxidizing agents. The products obtained through the photochemical oxidation process are shown to be biosurfactants with chemical compositions and properties close to those of the highest-performance rhamnolipids and sophorolipids bacterial surfactants. Due to the wide use of surfactants in the chemical industry and particularly the high market value of bacterial surfactants, the obtained results lay the basis for a feasible implementation of the process at the industrial level and the realization of a biorefinery using not only MBW but also biowastes from other sources (e.g., agriculture and the agro-industry) as feedstock.

To obtain hints for further R&D work aiming to improve the composition and properties of the photo-oxidized biosurfactants, a comparative review of the composition, properties, and production cost of the chemical and bacterial surfactants has also been carried out in the present work. The results in Section 4.4 show that the photo-oxidized biosurfactants derived from the MBW anaerobic digestate may not reach the performance level of the bacterial surfactants. Yet, the former biosurfactants obtained in the present work, according to scheme 1, have much more chance to be marketed products at large scales due to their low production costs. Contextually, without a change in paradigm in the R&D work on rhamnolipids and sophorolypids from biotechnology to integrated cheap biochemical–chemical processes, the bacterial surfactants may remain niche products.

Aside from the authors’ opinion on the above change in paradigm, the present proof-of-concept study introduces a highly sustainable chemical process for the production of biosurfactants from renewable biowastes and does not claim quality better than this. It is unquestionable that further R&D work on the integrated chemical–biochemical paradigm proposed in the present work for the implementation of SBP at the industrial level should answer important questions, as, for example, (i) what are, and how can be identified and/or isolated from the SBP heterogeneous molecular pool the molecules performing as active principles in surfactant-based consumers’ finished products? (ii) Can the presence of potentially harmful molecules in the SBP molecular pool be excluded? (iii) What is the stability (durability) of the obtained products in the application context? (iv) How can the biowaste type and site-specific compositional variability affect the SBP composition, property, and performance? For SBP derived from different biowastes, and used as biosurfactants [10], biocatalysts [71], and agrochemicals and animal feed supplements [78], the results of previous work have evidenced no critical stability problems over 12 months storage in the dry solid state or in solution kept in the dark, nor toxicity or performance effects. However, the identification and/or isolation from the SBP heterogeneous molecular pool of molecules performing as active principles for specific tailored applications is likely to take very long and costly R&D work. The research risk in this case lies in the obtainment of low-value results relative to the amount of applied effort or in the production of high-performance surfactants for niche applications that are too expensive for use in place of synthetic surfactants for the manufacture of consumers’ products with high market turnover.

## Figures and Tables

**Figure 1 polymers-16-01479-f001:**
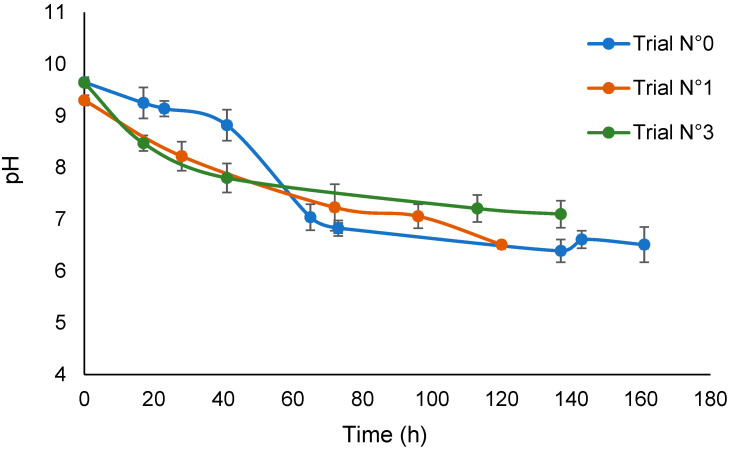
Plot of pH versus time for trials No.0, No.1, and No.3.

**Figure 2 polymers-16-01479-f002:**
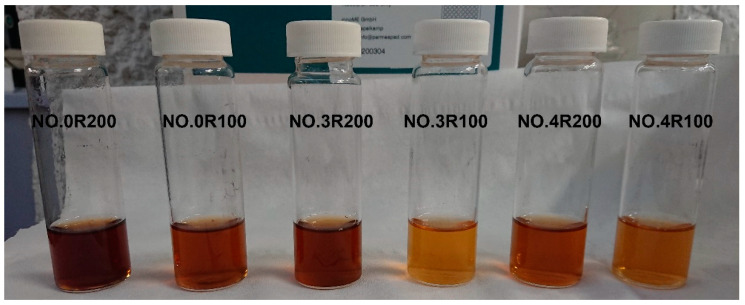
Color of the 2 g/L solutions of products identified by same treatment No. and corresponding Ri in Table 4.

**Figure 3 polymers-16-01479-f003:**
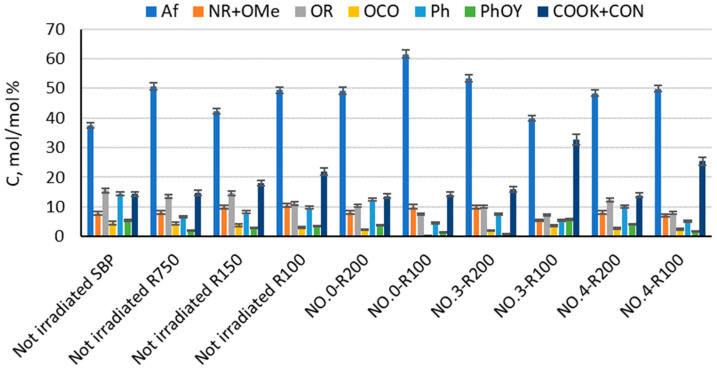
C types and functional groups relative composition as C mol/mol % and relative to total organic C for pristine SBP and products obtained according to the treatments identified by the abbreviations in Table 1, Table 2 and Table 3, i.e., the fractions (Ri, i = 750, 200, 150, and 100) with molecular weights ≥ 100 kDa obtained by sequential membrane ultrafiltration of the crude insoluble products at pH < 1. Legends: aliphatic C (Af), amine and methoxy C (NR + OMe), alkoxy C (OR), anomeric C (OCO), aromatic C (Ph), phenol and phenoxy C (PhOY), carboxylate and amide C (COOK + CON), R = alkyl, and Y = H, R, and Ar.

**Figure 4 polymers-16-01479-f004:**
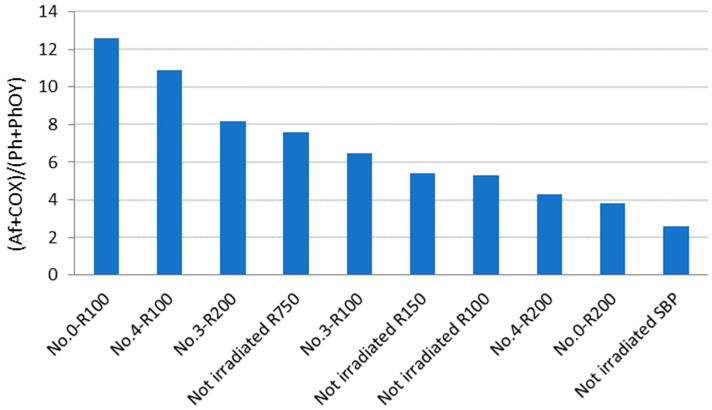
Total aliphatic and carboxyl C moles per mole of total aromatic C for products in Figure 3 and Appendix A.

**Figure 5 polymers-16-01479-f005:**
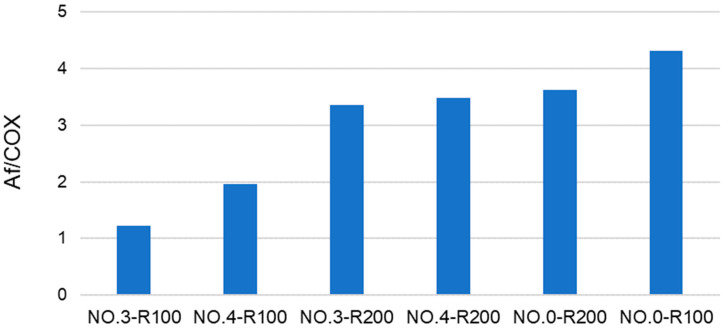
Aliphatic/carboxyl (Af/COX) C ratio for products obtained by the irradiation of pristine SBP (See Figure 3 and Appendix A).

**Figure 6 polymers-16-01479-f006:**
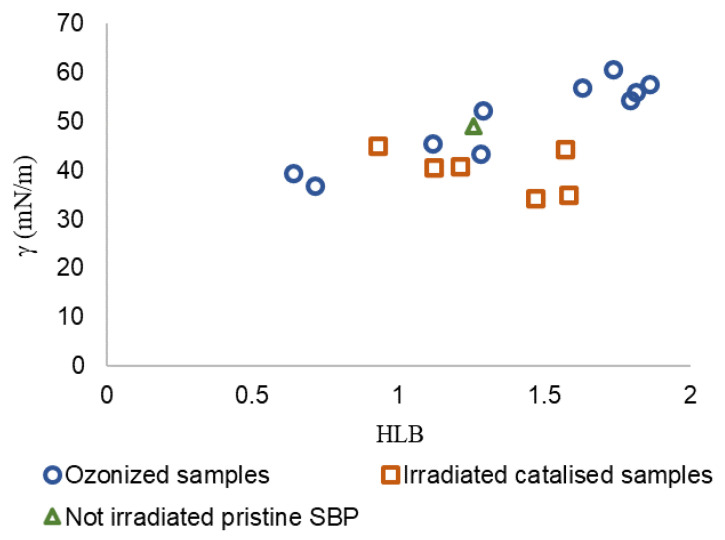
Scatter plot of data for irradiated catalyzed samples (No.0, No.3, and No.4 listed in Table 3) and of data reported in previous work [12] for not-irradiated crude pristine SBP and the 10 fractions with molecular weights in the 30–750 kDa range obtained by ozonization of pristine SBP at different reaction times.

**Figure 7 polymers-16-01479-f007:**
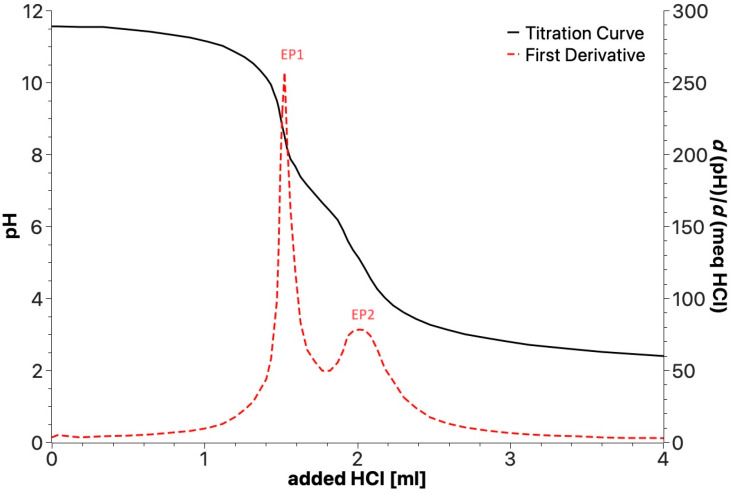
Example of potentiometric titration curve for products obtained by irradiation of SBP.

**Figure 8 polymers-16-01479-f008:**
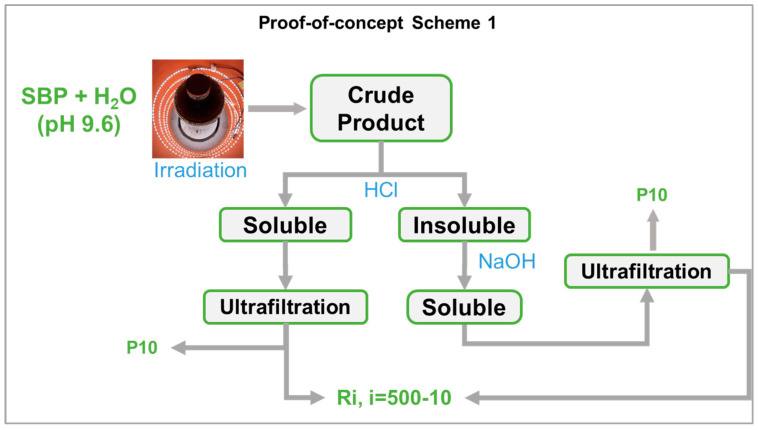
Representation of the experimental conditions described in Section 2 to produce the crude products by simulated solar light irradiation and isolation of the fractions listed in Table 4 for the treatments No.0, No.3, and No.4.

**Figure 9 polymers-16-01479-f009:**
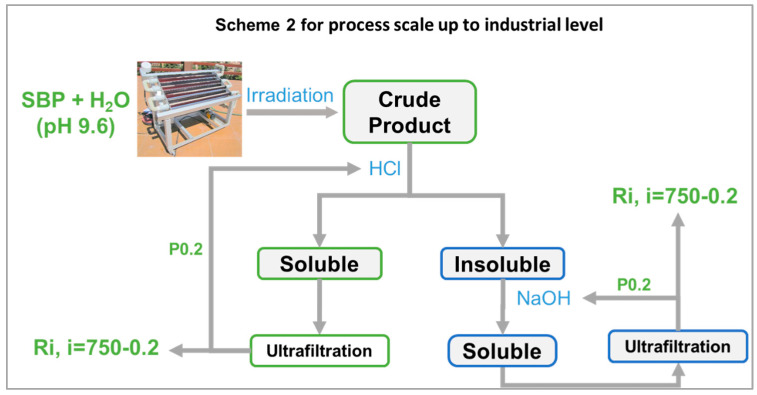
Feasible scale-up to industrial level of process depicted in Figure 8 based on solar light irradiation of SBP, membrane ultrafiltration to isolate retentates (Ri) with different molecular weights in the ≥750 kDa–≥0.2 kDa range, and recovery and recirculation of added mineral reagents contained in the permeate fraction (P0.2) with a molecular weight of <0.2 kDa.

**Figure 10 polymers-16-01479-f010:**
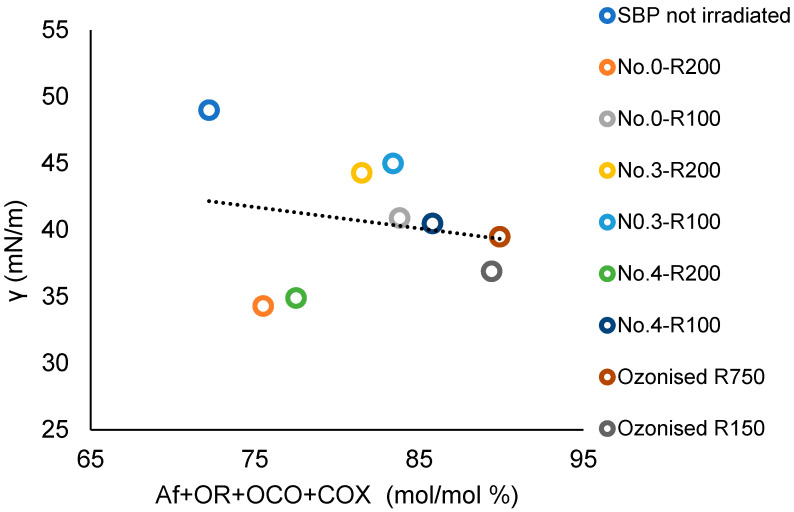
Plot of the surface tension (γ) at 2 g/L concentration of not-irradiated, irradiated, and ozonized SBP products versus the total Af + OR + OCO + COX mol/mol % carbon content in each product.

**Figure 11 polymers-16-01479-f011:**
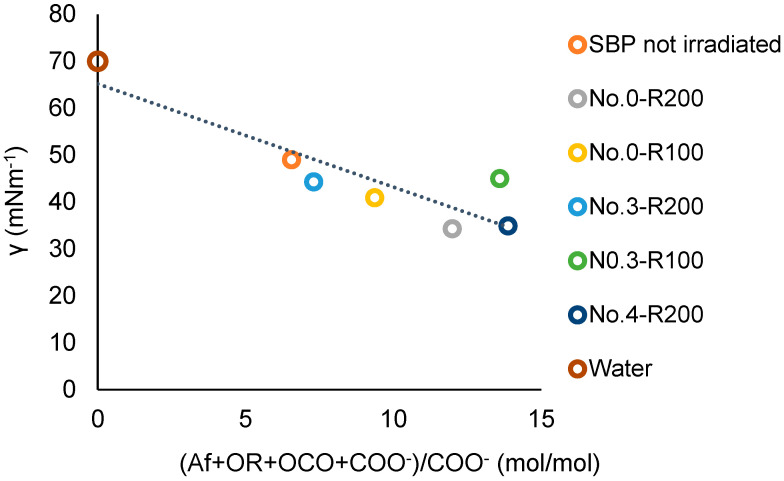
Plot of the surface tension (γ) at 2 g/L concentration of not-irradiated and irradiated SBP products versus the total Af + OR + OCO + COO^−^)/COO^−^ mol/mol carbon content ration in each product.

**Figure 12 polymers-16-01479-f012:**
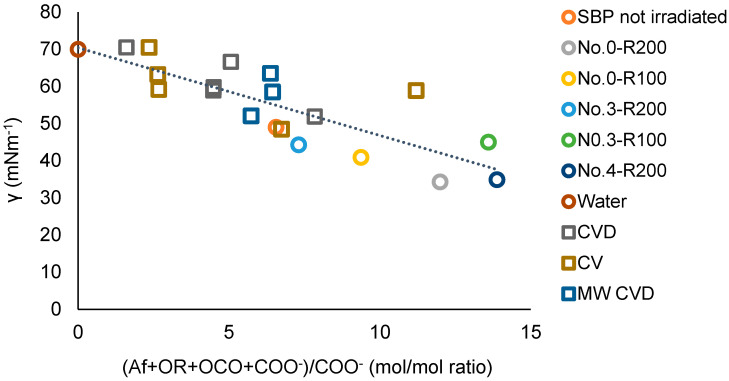
Plot of the surface tension (γ) at 2 g/L concentration of not-irradiated and irradiated SBP, ozonized SBPs from CV and CVD composts [13], and SBPs from the hydrolysis of MW CVD compost at 60–100 °C [14] versus the total Af + OR + OCO + COO^−^)/COO^−^ mol/mol carbon content ratio in each product. See also Appendix A.

**Table 1 polymers-16-01479-t001:** Experimental conditions of SBP treatments by irradiation of SBP solution in the absence and presence of added hydrogen peroxide at 3 × 10^−5^ H_2_O_2_/C mol/mol ratio and of added KOH to keep the starting pH constant.

Treatment No.	H_2_O_2_	KOH	Irradiation Days	pH	
				Start	End
None ^a^	no	no	0		
0	no	no	7	9.6	6.5
1	yes	no	5	9.8	6.5
2	yes	yes	7	9.8	9.8
	yes	no	+12	9.8	7.1
	yes	no	+9	7.1	7.0
*Total irradiation days for No.2*			28		
3	yes	no	6	9.6	7.1
4	yes	yes	6	9.7	9.7

^a^ SBP solution before irradiation.

**Table 2 polymers-16-01479-t002:** Mass and C balance for the acidification treatment of the not-irradiated and irradiated solutions of SBP, according to Table 1 experimental conditions.

Post-Treatment No. ^a^	Mass w/w % and C mol/mol % Yields, Relative to Pristine SBP, for Insoluble and Soluble Matter in Water at pH < 1			
	Insoluble Matter		Soluble Matter	
	Mass	C	Mass	C
None	65 ± 7	71 ± 7	30 ± 6	9.4 ± 3
0	48 ± 1	63 ± 4	53 ± 8	20 ± 2
1	28 ± 4	38 ± 2	n.d. ^b^	n.d. ^b^
2	35 ± 5	47 ± 5	n.d. ^b^	n.d. ^b^
3	35 ± 2	46 ± 3	61 ± 4	42 ± 1
4	37 ± 3	48 ± 6	61 ± 9	30 ± 3

^a^ Identified by the same No. in Table 1. ^b^ not determined.

**Table 3 polymers-16-01479-t003:** Molecular weight distribution for soluble and insoluble matter in Table 2. Mass weight w/w %, relative to pristine SBP, recovered with major fractions obtained by membrane ultrafiltration of the insoluble and soluble products in acid water at pH < 1.

Treatment No. ^a^	Product	R750	R150	R100	R5	P0.2		Total ^b^
None (not-irradiated SBP)	crude	58 ± 3	20 ± 2	5 ± 2	2 ± 1	15 ± 4		100
		**R500**	**R200**	**R100**	**R50**	**R10**	**P10**	
0	insoluble	8.7 ± 1.6	21.7 ± 4	7.4 ± 3.1			7.9 ± 4.2	45.7
	soluble	1.9 ± 0.6		5.0 ± 2.7		22.8 ± 5	23.7 ± 5	53.4
3	insoluble	2.7 ± 0.4	29.8 ± 7	4.3 ± 2.3	0.2 ± 0.1	0.9 ± 0.2	3.7 ± 1.8	41.6
	soluble	1.9 ± 0.8		13.4 ± 3	3 ± 0.4	33.4 ± 7	14 ± 2	65.7
4	insoluble	1.5 ± 0.6	31.6 ± 6	6.1 ± 2		0.3 ± 0.1	4.3 ± 0.2	43.8
	soluble	1.5 ± 0.4	1.1 ± 0.5	17.8 ± 6	1.9 ± 0.2		36.8 ± 8	59.1

^a^ Identified by the same No. in Table 1 and Table 2. ^b^ Total weight of insoluble and soluble matter and fractions isolated in each treatment. No insoluble matter in the pristine not-irradiated SBP.

**Table 4 polymers-16-01479-t004:** Surface tension (**γ**, mN/m) for fraction (Ri) insoluble at acid pH obtained by membrane ultrafiltration of crude products obtained in trials No. 0, 3, and 4 (Table 3) and of not-irradiated pristine and ozonized SBP [12].

Treatment No./Products	Ri	γ (2 g/L)	CMC (g/L)	γ_cmc_
0	R200	34.3 ± 0.0	0.4	39
	R100	40.9 ± 1.3	0.5	46
3	R200	44.3 ± 0.3	0.5	46
	R100	45.0 ± 1.8		
4	R200	34.5 ± 0.2	0.4	35
	R100	40.5 ± 0.9	0.5	48
Pristine SBP [12]	R750	56.8 ± 0.8		
	R150	57.3 ± 0.0		
	R100	54.2 ± 0.7		
Pristine SBP ozonized 64 h [12]	R750	39.5 ± 1.0		
	R150	36.8 ± 0.6	0.47	38
	R100	43.6 ± 2.3		

**Table 5 polymers-16-01479-t005:** Group contribution numbers for different structural units containing carboxylic, amide, and amine functional groups.

Structural Unit	Group-Contribution Number [31]
-CONR_2_	1.9–2.7
-CH(NH_3_)^+^-COO^−^	4.3
-COO^−^	12.7–21
-COO-ester	1.1–2.3
-COOH	2.09
-NR_2_, -NR_2_^+^	7.0–9.4

**Table 6 polymers-16-01479-t006:** Content (meq/g dry matter) ^a^ of phenol PhOH and carboxylate (-COOH I and -COOH II) functional groups in the not-irradiated SBP and in the retentates (R200 and R100) isolated by membrane ultrafiltration of the irradiated crude SBP products.

Sample	PhOH	-COOH I	-COOH II	Total -COOH
SBP	0.72 ± 0.10	2.11 ± 0.11	0.97 ± 0.17	3.07 ± 0.18
No.0-R200	0.48 ± 0.09	1.76 ± 0.14	0.40 ± 0.03	2.15 ± 0.15
No.0-R100	0.73 ± 0.34	1.86 ± 0.25	0.50 ± 0.01	2.35 ± 0.26
No.3-R200	0.25 ± 0.00	1.61 ± 0.08	0.30 ± 0.07	1.91 ± 0.00
No.3-R100	0.49 ± 0.10	1.86 ± 0.02	0.48 ± 0.05	2.34 ± 0.07
No.4-R200	0.40 ± 0.09	1.61 ± 0.08	0.32 ± 0.06	1.92 ± 0.01

^a^ Mean ± standard deviation values calculated from three replicate measurements.

**Table 7 polymers-16-01479-t007:** C types and functional groups content (mol/mol %) ^a^ for products in Table 6.

	SBP	No.0-R200	No.0-R100	No.3-R200	No.3-R100	No.4-R200
Af	37.6 ± 1.1	49.2 ± 1.2	61.6 ± 2.7	53.4 ± 2.9	39.8 ± 0.9	48.4 ± 1.1
NR + OMe	7.9 ± 0.2	8.1 ± 0.5	10.1 ± 0.4	10 ± 0.5	5.5 ± 0.8	8.2 ± 0.5
OR	15.6 ± 0.4	10.4 ± 0.8	7.5 ± 0.3	10.1 ± 0.6	7.3 ± 1.1	12.4 ± 0.4
OCO	4.6 ± 0.1	2.3 ± 0.3	0.4 ± 0.0	2.1 ± 0.1	3.7 ± 0.5	2.8 ± 0.2
Ph	14.4 ± 0.2	12.5 ± 0.7	4.6 ± 0.3	7.6 ± 0.4	5.4 ± 0.8	10.2 ± 0.3
PhOH	2.3 ± 0.1	1.3 ± 0.1	1.2 ± 0.2	0.7 ± 0.1	2.2 ± 0.2	1.0 ± 0.1
PhOR	3.1 ± 0.1	2.6 ± 0.1	0.2 ± 0.1	0.2 ± 0.1	3.5 ± 0.2	3.2 ± 0.0
COO^−^	10.4 ± 0.3	5.6 ± 0.2	8.3 ± 0.1	5.2 ± 0.1	10.5 ± 1.0	4.9 ± 0.1
CONR_2_	4.0 ± 0.3	8.0 ± 0.2	6.1 ± 0.6	10.7 ± 0.9	22.1 ± 1.8	9.0 ± 1.4
C ^b^	2.98	3.73	2.89	3.66	2.23	3.89

^a^ Values referred to the total organic C moles in the product. COO^−^ values correspond to the sum of the COOH I and COOH II values reported in Table 6. R stands for alkyl or aryl C moiety. ^b^ C mol/w % referred to the product dry matter measured by microelemental analysis used to convert meq/g potentiometric data in Table 6 into Table 7 mol/mol % values.

## Data Availability

Data are contained within the article and Appendix A.

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
