# Peer review of "A Low-Cost Ecofriendly Oxidation Process to Manufacture High-Performance Polymeric Biosurfactants Derived from Municipal Biowaste"

_polymers, 2024, doi:10.3390/polym16111479_

Round 1

Reviewer 1 Report

Comments and Suggestions for Authors

The manuscript deals with the resource and cost efficient production of biosurfactants from fermented municipal biowaste that is size-purified and treated photocatalytically.  
The manuscript is overall well written and logically structures. The motivation is clear and the used methods appear to be well established. Also the data interpretation appears to be justified. What remains unclear to me, even after quite some chemical analysis, what kind of biosurfactants are left in the product and how much they are chemically transformed into something undefined by the treatment. How can the authors exclude that the composition of the product is not merely a wild chemical mixture with surfactant character? How can they exclude that some constituents of the product are harmful for the processes that require surfactants and would otherwise rely on non-biological surfactants or expensive biosurfactants?
This aspect should be explained more clearly.

Author Response

see file rev1

Reviewer 2 Report

Comments and Suggestions for Authors

Building on previous experience Baglieri et al. report here on a “A low cost ecofriendly oxidation process to manufacture high performance polymeric biosurfactants derived from municipal biowaste”. In this proof-of concept study the authors introduce a more sustainable chemical oxidation process in warter for the production biosurfactants from renewable municipal biowaste. The evaluation of the feasibility of scaling-up is very important as is the attempt to correlate the properties of the recovered material with their functional characteristics and improve them. 

While not claiming quality better than this of the convention industrial process, this work proposes a cheap alternative that could, according to the evaluation presented in the manuscript, be scaled up.

It is a very interesting manuscript which was a pleasure to read. The experiments and results are carefully described, presented and evaluated. The authors are to be commented for a very interesting and critical discussion and evaluation of  the optimization and implementation of this approach. The claims of this study will certainly be of interest to the readership of the journal.

Few Comments:

1.    In Figure 1 a plot of pH versus time for trials No.0, No.1 and No.3 is presented. In this plot, trials 1 and 3 are depicted together. However in table 1 the pH values are different while, as the authors, state the total acid production is the same. The graph should be corrected to represent all 3 trials (and error bars, vide supra).

2.    Importantly, no errors/error bars are included in any of the tables of graphs. Were the experiments performed only one time? This is the most important criticism to this work: at least triplicates should be performed to substantiate and fully support the very interesting findings of this study. 

3.    The solubility and molecular weight of the untreated and treated SBP products is evaluated upon lowering the pH to 1. Interesting trends are observed. The question raised is whether these harsh conditions affect the composition of SBPs. Has this been tested?

4.    Many of the measurements presented in the manuscript were performed after membrane ultrafiltration. Was the total mass of samples before and after ultrafiltration evaluated? Does this methodology have limitations? If so, they should be discussed in the manuscript. 

5.    The CPMAS provided very interesting results. Again, no error bars are included. Please add a reference for CPMAS quantification. 

Few Typos

Line 241: 750 kDa to 0.2 kD.  

Line 439: ammine

Line 514-516: These findings confirm previous knowledge that, although oxidation of carbon atoms into CO2 is the most thermodynamically stable carbon compound, the mineralisation of organic molecules is markedly slower than their de-aromatisation …” Please add a “for” after although ore rephrase

To conclude, it was a pleasure to review this manuscript and I will be very happy to support its publication once these comments are addressed.

Author Response

see file Rev 2

Reviewer 3 Report

Comments and Suggestions for Authors

The article has a very specific structure: It is a combination of a research article and a review article (section 4.4). I think that for greater clarity of the paper, the authors should separate these two aspects. A consequence of this work arrangement are also conclusions that do not directly refer to the obtained research results. The conclusions contained are very general and constitute discussions on the potential for the implementation of the research obtained on an industrial scale. But what exactly did these studies achieve? Which variant is the most advantageous from the point of view of the functional properties of the obtained product?

Detailed comments:

1) How many repetitions were the experiments carried out? How is the repeatability of properties in subsequent experiments?

2) What is the stability (durability) of the obtained products in the application context?

3) There is no description of how the membrane process was carried out (flow conditions - crossflow, dead-end; equipment; membranes – producer; volumes of solutions: feed, permeate, and retentate).

4) Fig. 1 - Is the course of pH for trials No. 1-3 exactly the same over time (7 days of irradiation)?

5) The values in Table 3 (No. 0, 3 and 4) do not balance.

6) How was the cmc value tested?

Author Response

see file Rev 3

Round 2

Reviewer 3 Report

Comments and Suggestions for Authors The most important comments have been taken into account. The article may be published in its current form.